# Towards Model-Based Contrastive Explanations for Explainable Planning

**Benjamin Krarup**[*]**, Michael Cashmore**[*]**, Daniele Magazzeni**[*]**, Tim Miller**[†]

### Abstract

An important type of question that arises in Explainable Planning is a contrastive question, of the form "Why action A instead of action B?". These kinds of questions can be answered with a *contrastive explanation* that compares properties of the original plan containing A against the contrastive plan containing B. An effective explanation of this type serves to highlight the differences between the decisions that have been made by the planner and what the user would expect, as well as to provide further insight into the model and the planning process. Producing this kind of explanation requires the generation of the contrastive plan. This paper introduces domain-independent compilations of user questions into constraints. These constraints are added to the planning model, so that a solution to the new model represents the contrastive plan. We introduce a formal description of the compilation from user question to constraints in a temporal and numeric PDDL2.1 planning setting.

## 1 Introduction

Explainable AI (XAI) is an emerging and important research area within AI. Recent work has shown that AI Planning is an important tool in XAI, as its decision-making mechanisms are model-based and so in principle more transparent. This recent work includes many approaches towards providing explanations in AI planning.

Chakraborti et al. (2019) gives an in-depth overview of this work and different terms used within the XAI landscape. In particular, Zhang et al. (2017) shows that if an AI system behaves "explicably" there is less of a need for explanations. However, this is not always possible and explanation is sometimes required. Chakraborti et al. (2017) tackles explanation as a model reconciliation problem, arguing that the explanation must be a difference between the human model and AI model. Seegebarth et al. (2012) show that by representing plans as first order logic formulae generating explanations is feasible in real time. In contrast, in this paper we focus on contrastive *"why"* questions. Fox, Long, and Magazzeni (2017) highlight some important questions in XAIP and discuss possible answers, and also describe how these *"why"* questions are especially important. Smith (2012) outlines the approach to planning as an iterative process for bet-

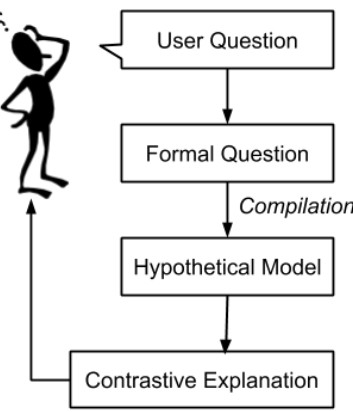

Figure 1: The four-stage process for generating a contrastive explanation from a user question. The hypothetical model is created by compiling the formal question into the planning model (in PDDL 2.1).

ter modelling preferences and providing explanations. We propose to follow this same approach.

The aim of explanations is to improve the user's levels of understanding and trust in the system they are using. These explanations can be local (regarding a specific plan) or global (concerning how the planning system works in general). In this paper we focus on local explanations of temporal and numeric planning problems, introducing an approach for explaining why a planner has made a certain decision. Through active exploration of these specific cases, the user may also gain global insight into the way in which the planner makes decisions. (See (Lipton 1990; 2016; Ribeiro, Singh, and Guestrin 2016)).

To achieve an understanding of a decision, it is important that explanations adapt to the specific context and mental model of the user. One step towards this is to support the user iteratively asking different questions suitable for their context. Haynes et al. (2009) identify ten question types that a user might have about an intelligent system, also described by Mueller et al. (2019). Lim et al. (2009) show in a grounded study that of these, the questions *why* and *why not* provided the most benefit in terms of objective understanding and feelings of trust. In the context of planning *why not*

---

[*]King's College London, UK, {*firstname.lastname*}*@kcl.ac.uk*

[†]University of Melbourne, Australia, *tmiller@unimelb.edu.au*

questions are *contrastive questions*, because the user is asking why some action was selected rather than some other action that was not.

Instead, Miller argues that all such questions can be asked as contrastive questions of the form "Why action A rather than action B?" (Miller 2018). Contrastive questions capture the context of the question; they more precisely identify the gaps in the user's understanding of a plan that needs to be explained (Lewis 1986). A contrastive question about a plan can be answered by a *contrastive explanation*. Contrastive explanations will compare the original plan against a contrastive plan that accounts for the user expectation. Providing contrastive explanations is not only effective in improving understanding, but is simpler than providing a full causal analysis (Miller 2019).

Following the approach of Smith (2012) we propose an approach to contrastive explanations through a dialogue with the user. The proposed approach consists of an iterative four-stage process illustrated in Figure 1. First the user asks a contrastive question in natural language. Second, a constraint is derived from the user question, in the following we refer to this constraint as the *formal question*. Third a hypothetical model (HModel) is generated which encapsulates this constraint. A solution to this model is the hypothetical plan (HPlan) that can be compared to the original plan to show the consequence of the user suggestion. The user can compare plans and iterate the process by asking further questions, and refining the HModel. This allows the user to combine different compilations to create a more constrained HModel, producing more meaningful explanations, until the explanation is satisfactory. Each stage of this process represents a vital research challenge. This paper describes and formalises the third stage of this process: compiling the formal question into a hypothetical model for temporal and numeric planning.

We are interested in temporal and numeric planning problems, for which optimal solutions are difficult to find. Therefore, while the process described above serves for explanation, the insight of the user can also result in guiding the planning process to a more efficient solution. As noted by (Smith 2012), the explanations could also give the user the opportunity to improve the plan with respect to their own preferences. The user could have hidden preferences which have not been captured in the model. The user could ask questions which enforce constraints that favour these preferences. The new plan could be sub-optimal, but more preferable to the user.

The contribution of this paper is a formalisation of domain-independent and planner-agnostic compilations from formal contrastive questions to PDDL2.1 (Fox and Long 2003), necessary for providing contrastive explanations. The compilations shown are not exhaustive. However, they do cover an interesting set of questions which users would commonly have about both classical and temporal plans. The paper is organised as follows. The next section describes the planning definitions we will use throughout the paper. In Section 3 we describe the running example that we use to demonstrate our compilations throughout the paper. In Section 4 we list the set of formal questions that we are interested in, and formalise the compilations of each of these into constraints. Finally, we conclude the paper in Section 5 whilst touching on some interesting future work.

## 2 Background

Our definition of a planning model follows the definition of PDDL2.1 given by (Fox and Long 2003), extended by a set of time windows as follows.

**Definition 1** *A **planning model** is a pair $\Pi = \langle D, Prob \rangle$. The domain $D = \langle Ps, Vs, As, arity \rangle$ is a tuple where $Ps$ is a finite set of predicate symbols, $Vs$ is a finite set of function symbols, $As$ is a set of action schemas, called operators, and $arity$ is a function mapping all of these symbols to their respective arity. The problem $Prob = \langle Os, I, G, W \rangle$ is a tuple where $Os$ is the set of objects in the planning instance, $I$ is the initial state, $G$ is the goal condition, and $W$ is a set of time windows.*

A set of atomic *propositions* $P$ is formed by applying the predicate symbols $Ps$ to the objects $Os$ (respecting arities). One proposition $p$ is formed by applying an ordered set of objects $o \subseteq O$ to one predicate $ps$, respecting its arity. For example, applying the predicate $(block\_on\,?a\,?b)$ with arity 2 to the ordered set of objects $\{blockA, blockB\}$ forms the proposition $(block\_on\,blockA\,blockB)$. This process is called "grounding" and is denoted with:

$$ground(ps, \chi) = p$$

where $\chi \subseteq O$ is an ordered set of objects. Similarly the set of *primitive numeric expressions* (PNEs) $V$ are formed by applying the function symbols $Vs$ to $Os$.

A state $s$ consists of a time $t \in \mathbb{R}$, a logical part $s_l \subseteq P$, and a numeric part $s_v$ that describes the values for the PNE's at that state. The initial state $I$ is the state at time $t = 0$.

The goal $G = g_1, ..., g_n$ is a set of constraints over $P$ and $V$ that must hold at the end of an action sequence for a plan to be valid. More specifically, for an action sequence $\phi = \langle a_1, a_2, \ldots, a_n \rangle$ each with a respective time denoted by $Dispatch(a_i)$, we use the definition of plan validity from (Fox and Long 2003) (Definition 15 "Validity of a Simple Plan"). A simple plan is the sequence of actions $\phi$ which defines a happening sequence, $t_{i=0\ldots k}$ and a sequence of states, $s_{i=0\ldots k+1}$ such that $s_0 = I$ and for each $i = 0 \ldots k$, $s_{i+1}$ is the result of executing the happening at time $t_i$. The simple plan $\phi$ is valid if $s_{k+1} \models G$.

Each time window $w \in W$ is a tuple $w = \langle w_{lb}, w_{ub}, w_v \rangle$ where $w_v$ is a proposition which becomes true or a numeric effect which acts upon some $n \in V$. $w_{lb} \in \mathbb{R}$ is the time at which the proposition becomes true, or the numeric effect is applied. $w_{ub} \in \mathbb{R}$ is the time at which the proposition becomes false. The constraint $w_{lb} < w_{ub}$ must hold. Note that the numeric effect is not effected at $w_{ub}$.

Similar to propositions and PNEs, the set of ground actions $A$ is generated from the substitution of objects for operator parameters with respect to it's arity. Each ground action is defined as follows:

**Definition 2** *A **ground action** $a \in A$ has a duration $Dur(a)$ which constrains the length of time that must*

```
(define (domain turtlebot_demo)
(:types waypoint robot)
(:predicates
  (robot_at ?v - robot ?wp - waypoint)
  (connected ?from ?to - waypoint)
  (visited ?wp - waypoint))
(:functions
  (travel_time ?wp1 ?wp2 - waypoint))
(:durative-action goto_waypoint
  :parameters (?v - robot
    ?from ?to - waypoint)
  :duration(= ?duration
    (travel_time ?from ?to))
  :condition (and
    (at start (robot_at ?v ?from))
    (over all (connected ?from ?to)))
  :effect (and
    (at start (not (robot_at ?v ?from)))
    (at end (visited ?to))
    (at end (robot_at ?v ?to)))))
```

Figure 2: The robotics domain used as a running example.

```
(define (problem task)
(:domain turtlebot_demo)
(:objects
    wp0 wp1 wp2 wp3 wp4 wp5 - waypoint
    kenny - robot)
(:init
    (robot_at kenny wp0) (visited wp0)
    (connected wp0 wp2) (connected wp0 wp4)
    (connected wp1 wp0) (connected wp1 wp2)
    (connected wp2 wp1) (connected wp2 wp4)
    (connected wp2 wp5) (connected wp3 wp5)
    (connected wp5 wp0) (connected wp5 wp2)
    (connected wp5 wp3)
    (= (travel_time wp0 wp2) 1.45)
    (= (travel_time wp0 wp4) 2)
    ...
(:goal (and (visited wp1) (visited wp2)
  (visited wp3)  (visited wp4) (visited wp5)
  )))
```

Figure 3: Example Problem with some travel time functions omitted for space.

*pass between the start and end of a; a start (end) condition $Pre_{\vdash}(a)$ ($Pre_{\dashv}(a)$) which must hold at the state that a starts (ends); an invariant condition $Pre_{\leftrightarrow}(a)$ which must hold throughout the entire execution of a; add effects $Eff(a)_{\vdash}^{+}$, $Eff(a)_{\dashv}^{+} \subseteq P$ that are made true at the start and ends of the action respectively; delete effects $Eff(a)_{\vdash}^{-}$, $Eff(a)_{\dashv}^{-} \subseteq P$ that are made false at the start and end of the action respectively; and numeric effects $Eff(a)_{\vdash}^{n}$, $Eff(a)_{\leftrightarrow}^{n}$, $Eff(a)_{\dashv}^{n}$ that act upon some $n \in V$.*

## 3 Running Example

We use as a running example the following planning model. Figure 2 shows the domain $D$. The domain describes a scenario in which a robot is able to move between connected waypoints and mark them as visited. The domain contains three predicate symbols ($robot\_at$, $connected$, $visited$) with arities 2, 2, and 1 respectively. The domain includes only a single function symbol $travel\_time$ with arity 2. There is a single operator $goto\_waypoint$.

Figure 3 shows the problem $Prob$. The problem specifies 7 objects: $wp0, wp1, wp2, wp3, wp4, wp5$ and $kenny$. The initial state specifies which propositions are initially true, such as the current location of the robot ($robot\_at\ kenny\ wp0$), and the initial values of the PNEs, e.g. ($= (travel\_time\ wp5\ wp3)\ 4.68$). The goal is specified as a constraint over $P \cup V$, in this example it is that the robot has visited all of the locations.

Figure 5 shows an example plan that solves this problem. This plan might appear sub-optimal. The robot moves from waypoint $wp2$ to $wp1$ and then immediately returns to $wp2$. This second action might seem redundant to the user. However, upon closer inspection of the connectivity of waypoints (shown in Figure 4) we can see that the plan is in fact the optimal one. Visiting waypoint $wp1$ is a goal of the problem, and it is only connected to waypoints $wp0$ and $wp2$, both of which have already been visited. Waypoint $wp0$ is only

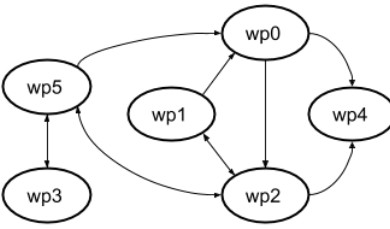

Figure 4: Waypoint connectivity in the running example. The robot is only allowed to move along the directed arrows.

```
0.00: (goto_waypoint kenny wp0 wp2)   [1.45]
1.45: (goto_waypoint kenny wp2 wp1)   [2.00]
3.45: (goto_waypoint kenny wp1 wp2)   [2.00]
5.45: (goto_waypoint kenny wp2 wp5)   [2.00]
7.45: (goto_waypoint kenny wp5 wp3)   [4.68]
12.13: (goto_waypoint kenny wp3 wp5)  [4.68]
16.81: (goto_waypoint kenny wp5 wp0)  [0.99]
17.80: (goto_waypoint kenny wp0 wp4)  [2.00]
```

Figure 5: Plan generated from the example domain and problem. The cost of the plan is its duration (19.80).

connected to waypoints $wp2$ and $wp4$, $wp2$ has been visited and $wp4$ is a dead end. For these reasons combined, the only logical option is to move back to $wp2$ after completing the goal of visiting $wp1$. This type of behaviour similarly happens between waypoints $wp3$ and $wp5$.

A graphical representation such as Figure 4 is not always available, and so even for this simple model and plan, deducing the reasoning behind the planned actions is not trivial. This is an example of where XAIP is useful. Using our proposed approach the user could have asked the question: "Why do we use the action ($goto\_waypoint\ kenny\ wp1\ wp2$), rather than not using

```
0.00:  (goto_waypoint kenny wp0 wp2)   [1.45]
1.45:  (goto_waypoint kenny wp2 wp5)   [2.00]
3.45:  (goto_waypoint kenny wp5 wp3)   [4.68]
8.13:  (goto_waypoint kenny wp3 wp5)   [4.68]
12.81: (goto_waypoint kenny wp5 wp2)   [2.00]
14.81: (goto_waypoint kenny wp2 wp1)   [2.00]
16.81: (goto_waypoint kenny wp1 wp0)   [2.00]
18.81: (goto_waypoint kenny wp0 wp4)   [2.00]
```

Figure 6: The hypothetical plan that accounts for the user's suggestion, avoiding the action of moving from $wp1$ to $wp2$. The cost of the plan is its duration (20.81).

it?". From this question we could generate a contrastive plan with this constraint enforced (shown in Figure 6). Comparing the actions and costs of the original and the new plan could shed light on why the action needed to be used. The user can carry on asking questions until they were satisfied.

## 4 Formalisation

**Definition 3** *An **explanation problem** is a tuple $E = \langle \Pi, \phi, Q \rangle$, in which $\Pi$ is a planning model (Definition 1), $\phi$ is the plan generated by the planner, and $Q$ is the specific question posed by the user. The problem is to provide insight that helps the user to answer question $Q$.*

In this paper, we assume that the user knows the model $\Pi$ and the plan $\phi$, so answers such as stating the goal of the problem will not increase their understanding. Given this, we propose the following set of questions, and provide a formal description for compilations of this set of formal questions of temporal plans:

1. Why is action $a$ used in state $s$, rather than action $b$? (Section 4.1)

2. Why is action $a$ not used in the plan, rather than being used? (Section 4.2)

3. Why is action $a$ used in the plan, rather than not being used? (Section 4.3)

4. Why is action $a$ used outside of time window $w$, rather than only being allowed within $w$? (Section 4.4)

5. Why is action $a$ not used in time window $w$, rather than being used within $w$? (Section 4.5)

6. Why is action $a$ used at time $t$, rather than at least some time $t'$ after/before $t$? (Section 4.6)

7. Why is action $a$ not performed before (after) action $b$, rather than $a$ being performed after (before) $b$? (Section 4.7)

These questions were derived by systematically assessing ways that counterfactual situations could occur in plans, and choosing those that would be useful over many applications. This is not an exhaustive list of possible constraints that can be enforced upon the original model, however, it does represent a list of questions that would be useful in specific contexts and applications.

Part of being able to answer these questions is the ability to reason about what would happen in the counterfactual cases. We approach this problem by generating plans for the

counterfactual cases via *compilations*. A compilation of a planning instance where the model is given by $\Pi$, and a question is given by $Q$ is shown as $Compilation(\Pi, Q) = \Pi'$ where:

$$\Pi' = \langle \langle Ps', Vs, As', arity' \rangle, \langle Os, I', G', W' \rangle \rangle$$

We call $\Pi'$ the *hypothetical model*, or HModel.

However, $\Pi'$ can also be used as the input model so that the user can iteratively ask questions about some model, i.e:

$$Compilation(Compilation(\Pi, Q), Q')$$

This allows the user to stack questions, further increasing their understanding of the plan through combining compilations. Combining compilations this way provides a much wider set of possible constraints.

After the HModel is formed, it is solved to give the HPlan. Any new operators that are used in the compilation to enforce some constraint are trivially renamed back to the original operators they represent. For each iteration of compilation the HPlan is validated against the original model $\Pi$.

### 4.1 Replacing an Action in a State

Given a plan $\phi$, a formal question $Q$ is asked of the form:

*Why is the operator $o$ with parameters $\chi$ used in state $s$, rather than the operator $n$ with parameters $\chi'$? where $o \neq n$ or $\chi \neq \chi'$*

For example, given the example plan in Figure 5 the user might ask:

"Why is $(goto\_waypoint\,kenny\,wp2\,wp5)$ used, rather than $(goto\_waypoint\,kenny\,wp2\,wp4)$?"

They might ask this because a goal of the problem is to visit $wp4$. As the robot visits $wp5$ from $wp3$ later in the plan, it might make sense to the user for the robot to visit $wp4$ earlier, as $wp5$ will be visited at a later point.

To generate the HPlan, a compilation is formed such that the ground action $b = ground(n, \chi')$ appears in the plan in place of the action $a_i = ground(o, \chi)$. Given the example above $b = ground(goto\_waypoint, \{kenny, wp2, wp4\})$, and $a_i = ground(goto\_waypoint, \{kenny, wp2, wp5\})$. Given a plan:

$$\phi = \langle a_1, a_2, \ldots, a_n \rangle$$

The ground action $a_i$ at state $s$ is replaced with $b$, which is executed, resulting in state $I'$, which becomes the new initial state in the HModel. A time window is created for each durative action that is still executing in state $s$. These model the end effects of the concurrent actions. A plan is then generated from this new state with these new time windows for the original goal, which gives us the plan:

$$\phi' = \langle a_1', a_2', \ldots, a_n' \rangle$$

The HPlan is then the initial actions of the original plan $\phi$ concatenated with $b$ and the new plan $\phi'$:

$$\langle a_1, a_2, \ldots, a_{i-1}, b, a_1', a_2', \ldots, a_n' \rangle$$

Specifically, the HModel $\Pi'$ is:

$$\Pi' = \langle \langle Ps, Vs, As, arity \rangle, \langle Os, I', G, W \cup C \rangle \rangle$$

where:

- $I'$ is the final state obtained by executing[1] $\langle a_1, a_2, \ldots, a_{i-1}, b \rangle$ from state $I$.

- $C$ is a set of time windows $w_x$, for each durative action $a_j$ that is still executing in the state $I$. For each such action, $w_x$ specifies that the end effects of that action will become true at the time point at which the action is scheduled to complete. Specifically: $w_x = \langle Dispatch(a_j) + Dur(a_j) - Dispatch(b), inf, u \rangle$ where $u = \mathit{Eff}(a_j)^-_\dashv \cup \mathit{Eff}(a_j)^+_\dashv \cup \mathit{Eff}(a_j)^n_\dashv$.

In the case in which an action $a_j$ that is executing in state $I'$ has an overall condition that is violated, this is detected when the plan is validated against the original model. As an example, given the user question above, the new initial state $I'$ from the running example is shown below:

```
(:init
(robot_at kenny wp4)  (visited wp2)
(visited wp1) (visited wp4)
(connected wp0 wp2)  (connected wp0 wp4)
(connected wp1 wp0)  (connected wp1 wp2)
...)
(:goal (and (visited wp1)(visited wp2)
  (visited wp3)(visited wp4) (visited wp5)
  )))
```

This captures the state $I'$, resulting from executing the actions $a_1, a_2, a_3$, and $b$:

```
0.00: (goto_waypoint kenny wp0 wp2)  [1.45]
1.45: (goto_waypoint kenny wp2 wp1)  [2.00]
3.45: (goto_waypoint kenny wp1 wp2)  [2.00]
5.45: (goto_waypoint kenny wp2 wp4)  [2.00]
```

In this state the robot has visited the waypoints $wp2$, $wp1$, and $wp4$, and is currently at $wp4$. This new initial state is then used to plan for the original goals to get the plan $\phi'$, which, along with $b$ and $\phi$, gives the HPlan. However, the problem is unsolvable from this state as there are no connections from $wp4$ to any other waypoint. By applying the user's constraint, and showing there are no more applicable actions, it answers the above question: "because by doing this there is no way to complete the goals of the problem".

This compilation keeps the position of the replaced action in the plan, however, it may not be optimal. This is because we are only re-planning after the inserted action has been performed. The first half of the plan, because it was originally planned to support a different set of actions, may now be inefficient, as shown by Borgo, Cashmore, and Magazzeni (2018).

If the user instead wishes to replace the action without necessarily retaining its position in the plan, then the following constraints on adding and removing an action from the plan can be applied iteratively, as mentioned previously.

### 4.2 Add an Action to the Plan

Given a plan $\phi$, a formal question $Q$ is asked of the form:

*Why is the operator $o$ with parameters $\chi$ not used, rather than being used?*

---

[1] We use VAL to validate this execution. We use the add and delete effects of each action, at each happening (provided by VAL), up to the replacement action to compute $I'$.

For example, given the example plan in Figure 5 the user might ask:

"Why is $(goto\_waypoint\ kenny\ wp2\ wp4)$ not used, rather than being used?"

They might ask this because a goal of the problem is to visit $wp4$. As the robot is at $wp2$ early in the plan, and you can visit $wp4$ from $wp2$, it might make sense to the user for the robot to visit $wp4$ at that time.

To generate the HPlan, a compilation is formed such that the action $a = ground(o, \chi)$ must be applied for the plan to be valid. The compilation introduces a new predicate $has\_done\_a$, which represents which actions have been applied. Using this, the goal is extended to include that the user suggested action has been applied. The HModel $\Pi'$ is:

$$\Pi' = \langle \langle Ps', Vs, As', arity' \rangle, \langle Os, I, G', W \rangle \rangle$$

where

- $Ps' = Ps \cup \{has\_done\_a\}$
- $As' = \{o'\} \cup As \setminus \{o\}$
- $arity'(x) = arity(x), \forall x \in arity$
- $arity'(has\_done\_a) = arity'(o') = arity(o)$
- $G' = G \cup \{ground(has\_done\_a, \chi)\}$

where the new operator $o'$ extends $o$ with the add effect $has\_done\_a$ with corresponding parameters, i.e.

$$\mathit{Eff}^+_\dashv(o') = \mathit{Eff}^+_\dashv(o) \cup \{has\_done\_a\}$$

For example, given the user question above, the operator $goto\_waypoint$ from the running example is extended to $goto\_waypoint'$ with the additional add effect $has\_done\_a$:

```
(:durative-action goto_waypoint'
  :parameters (?v - robot
    ?from ?to - waypoint)
  :duration( = ?duration
    (travel_time ?from ?to))
  :condition (at start (robot_at ?v ?from)
          (over all (connected ?from ?to))
  :effect (and (at end (visited ?to))
    (at start (not (robot_at ?v ?from)))
    (at end (robot_at ?v ?to))
    (at end (has_done_goto_waypoint' ?v ?from
?to)))))
```

and the goal is extended to include the proposition: $(has\_done\_goto\_waypoint\ kenny\ wp2\ wp4)$.

### 4.3 Remove a Specific Grounded Action

Given a plan $\phi$, a formal question $Q$ is asked of the form:

*Why is the operator $o$ with parameters $\chi$ used, rather than not being used?*

For example, given the example plan in Figure 5 the user might ask:

"Why is $(goto\_waypoint\ kenny\ wp1\ wp2)$ used, rather than not being used?"

A user might ask this because the robot has already satisfied the goal to visit $wp2$ before this point with the action $(goto\_waypoint\,kenny\,wp0\,wp2)$. The user might think the second action $(goto\_waypoint\,kenny\,wp1\,wp2)$ seems redundant.

The specifics of the compilation is similar to the compilation in Section 4.2. The HModel is extended to introduce a new predicate $not\_done\_action$ which represents actions that have not yet been performed. The operator $o$ is extended with the new predicate as an additional delete effect. The initial state and goal are then extended to include the user selected grounding of $not\_done\_action$. Now, when the user selected action is performed it deletes the new goal and so invalidates the plan. This ensures the user suggested action is not performed.

For example, given the user question above, an HPlan is generated that does not include the action $(goto\_waypoint\,kenny\,wp1\,wp2)$, and is shown in Figure 6.

### 4.4 Forbid an Action Outside a Time Window

Given a plan $\phi$, a formal question $Q$ is asked of the form:

*Why is the operator o with parameters $\chi$ used outside of time $lb < t < ub$, rather than only being allowed within this time window?*

For example, given the example plan in Figure 5 the user might ask:

"Why is $(goto\_waypoint\,kenny\,wp0\,wp4)$ used outside of times 0 and 2, rather than being restricted to that time window?"

A user can ask this because the action $(goto\_waypoint\,kenny\,wp0\,wp4)$ is used at the end of the plan, the robot starts at $wp0$ and must visit $wp4$ to satisfy a goal. The user might think that satisfying this goal earlier in the plan will free up time for the robot to complete the other goals.

To generate the HPlan, the planning model is compiled such that the ground action $a = ground(o,\chi)$ can only be used between times $lb$ and $ub$. To do this, the original operator $o$ is replaced with two operators $o_a$ and $o_{\neg a}$, which extend $o$ with extra constraints.

Operator $o_{\neg a}$ replaces the original operator $o$ for all other actions $ground(o,\chi')$, where $\chi' \neq \chi$. The action $ground(o_{\neg a},\chi)$ cannot be used (this is enforced using the compilation for forbidding an action described in Section 4.3). Operator $o_a$ acts as the operator $o$ specifically for the action $a = ground(o,\chi)$, which has an added constraint that it can only be performed between $lb$ and $ub$. Specifically, the HModel $\Pi'$ is:

$$\Pi' = \langle\langle Ps', Vs, As', arity'\rangle, \langle Os, I', G', W'\rangle\rangle$$

where:

- $Ps' = Ps \cup \{can\_do\_a, not\_done\_a\}$
- $As' = \{o_a, o_{\neg a}\} \cup As \setminus \{o\}$
- $arity'(x) = arity(x), \forall x \in arity$

- $arity'(can\_do\_a) = arity'(not\_done\_a) = arity'(o_a) = arity'(o_{\neg a}) = arity(o)$
- $I' = I \cup \{ground(not\_done\_a, \chi)\}$
- $G' = G \cup \{ground(not\_done\_a, \chi)\}$
- $W' = W \cup \{\langle lb, ub, ground(can\_do\_a, \chi)\rangle\}$

where the new operators $o_{\neg a}$ and $o_a$ extend $o$ with the delete effect $not\_done\_a$ and the precondition $can\_do\_a$, respectively. i.e:

$$Eff^-_\vdash(o_{\neg a}) = Eff^-_\vdash(o) \cup \{not\_done\_a\}$$
$$Pre_\vdash(o_a) = Pre_\vdash(o) \cup \{can\_do\_a\}$$

As the proposition $ground(can\_do\_a, \chi)$ must be true for $ground(o_a, \chi)$ to be performed, this ensures that the action $a$ can only be performed within the times $lb$ and $ub$. Other actions from the same operator can still be applied at any time using the new operator $o_{\neg a}$. As in Section 4.3 we make sure the ground action $ground(o_{\neg a}, \chi)$ can never appear in the plan.

For example, given the user question above, the operator $goto\_waypoint$ from Figure 2 is extended to $o_{\neg a}$ and $o_a$ as shown below:

```
(:durative-action goto_waypoint_nota
:parameters (?v - robot
    ?from ?to - waypoint)
 :duration(= ?duration
    (travel_time ?from ?to))
 :condition (and
    (at start (robot_at ?v ?from))
    (over all (connected ?from ?to)))
 :effect (and
    (at end (visited ?to))
    (at start (not (robot_at ?v ?from)))
    (at end (robot_at ?v ?to))
    (at start (not (not_done_goto_waypoint ?v
?from ?to)))))
```

```
(:durative-action goto_waypoint_a
:parameters (?v - robot
    ?from ?to - waypoint)
 :duration(= ?duration
    (travel_time ?from ?to))
 :condition (and (at start
(can_do_goto_waypoint ?v ?from ?to))
    (at start (robot_at ?v ?from))
    (over all (connected ?from ?to)))
 :effect (and (at end (visited ?to))
    (at start (not (robot_at ?v ?from)))
    (at end (robot_at ?v ?to))))
```

The initial state is extended to include the proposition $(not\_done\_goto\_waypoint\,kenny\,wp0\,wp4)$ and the time window $\langle 0, 2, (can\_do\_goto\_waypoint\,kenny\,wp0\,wp4)\rangle$. This time window enforces that the proposition $(can\_do\_goto\_waypoint\,kenny\,wp0\,wp4)$ is true between times 0 and 2. The resulting HPlan is:

```
0.00: (goto_waypoint kenny wp0 wp2)  [1.45]
1.45: (goto_waypoint kenny wp2 wp1)  [2.00]
3.45: (goto_waypoint kenny wp1 wp2)  [2.00]
5.45: (goto_waypoint kenny wp2 wp5)  [2.00]
7.45: (goto_waypoint kenny wp5 wp3)  [4.69]
```

```
12.14: (goto_waypoint kenny wp3 wp5)  [4.69]
16.83: (goto_waypoint kenny wp5 wp2)  [2.00]
18.84: (goto_waypoint kenny wp2 wp4)  [2.98]
```

Following the user suggestion, the action is no longer applied outside of the time window, and in fact does not appear in the plan at all.

## 4.5 Add an Action Within a Time Window

Given a plan $\phi$, a formal question $Q$ is asked of the form:

*Why is the operator $o$ with parameters $\chi$ not used at time $lb < t < ub$, rather than being used in this time window?*

For example, given the example plan in Figure 5 the user might ask:

"Why is $(goto\_waypoint\,kenny\,wp0\,wp4)$ not used between times 0 and 2, rather than being used in this time window?"

The HPlan given in Section 4.4 shows the user that there is a better plan which does not have the action in this time window. However, the user may only be satisfied once they have seen a plan where the action is performed in their given time window. To allow this the action may have to appear in other parts of the plan as well.

This constraint differs from Section 4.4 in two ways: first the action is now forced to be applied in the time window, and second the action can be applied at other times in the plan. This constraint is useful in cases such as a robot that has a fuel level. As fuel is depleted when travelling between waypoints, the robot must refuel, possibly more than once. The user might ask "why does the robot not refuel between the times $x$ and $y$ (as well as the other times it refuels)?".

To generate the HPlan, the planning model is compiled such that the ground action $a = ground(o, \chi)$ is forced to be used between times $lb$ and $ub$, but can also appear at any other time. This is done using a combination of the compilation in Section 4.2 and a variation of the compilation in Section 4.4. Simply, the former ensures that new action $ground(o_a, \chi)$ must appear in the plan, and the latter ensures that the action can only be applied within the time window. The variation of the latter compilation is that the operator $o_{\neg a}$ is not included, and instead the original operator is kept in the domain. This allows the original action $a = ground(o, \chi)$ to be applied at other times in the plan. Given this, the HModel $\Pi'$ is:

$$\Pi' = \langle\langle Ps', Vs, As', arity'\rangle, \langle Os, I, G', W'\rangle\rangle$$

where:
- $Ps' = Ps \cup \{can\_do\_a, has\_done\_a\}$
- $As' = \{o_a\} \cup As$
- $arity'(x) = arity(x), \forall x \in arity$
- $arity'(can\_do\_a) = arity'(has\_done\_a)$
  $= arity'(o_a) = arity(o)$
- $G' = G \cup \{ground(has\_done\_a, \chi)\}$
- $W' = W \cup \{\langle lb, ub, ground(can\_do\_a, \chi)\rangle\}$

As $wp4$ is a dead end there is no valid HPlan following this suggestion.

## 4.6 Delay/Advance an Action

Given a plan $\phi$, a formal question $Q$ is asked of the form:

*Why is the operator $o$ with parameters $\chi$ used at time $t$, rather than at least some duration $t'$ after/before $t$?*

For example, given the example plan in Figure 5 the user might ask:

"Why is $(goto\_waypoint\,kenny\,wp2\,wp5)$ used at time $5.45$, rather than at least 4 seconds earlier?"

A user might ask this question in general because they expected an action to appear earlier or later in a plan. This could happen for a variety of reasons. In domains with resources that are depleted by specific actions, and are replenished by others, such as fuel for vehicles, these questions may arise often. A user might want an explanation for why a vehicle was refueled earlier or later than what was expected. In this case the refuel action can be delayed or advanced to answer this question.

For this particular example the user might want the action $(goto\_waypoint\,kenny\,wp2\,wp5)$ to be advanced nearer the start of the plan. The user might see that in the original plan the robot goes from $wp2$ to $wp1$ at time $1.45$ and then instantly goes back again. The user might think that a better action would be to go from $wp2$ to $wp5$ before this. The user might notice that $wp5$ is connected to more waypoints than $wp1$. Having these extra options might prevent redundant actions that revisit waypoints.

To generate the HPlan, the planning model is compiled such that the ground action $a = ground(o, \chi)$ is forced to be used in time window $w$ which is at least $t'$ before/after $t$. This compilation is an example of a combination of two other compilations: adding an action (in Section 4.2) and forbidding the action outside of a time window (in Section 4.4). The latter enforces that the action can only be applied within the user specified time window, while the former enforces that the action must be applied. The HModel $\Pi'$ is:

$$\Pi' = \langle\langle Ps', Vs, As', arity'\rangle, \langle Os, I, G', W'\rangle\rangle$$

where:
- $Ps' = Ps \cup \{can\_do\_a, not\_done\_a, has\_done\_a\}$
- $As' = \{o_a, o_{\neg a}\} \cup As \setminus \{o\}$
- $arity'(x) = arity(x), \forall x \in arity$
- $arity'(can\_do\_a) = arity'(not\_done\_a) = arity'(has\_done\_a) = arity'(o_a) = arity'(o_{\neg a}) = arity(o)$
- $I' = I \cup \{ground(not\_done\_a, \chi)\}$
- $G' = G \cup \left\{ \begin{array}{l} ground(not\_done\_a, \chi), \\ ground(has\_done\_a, \chi) \end{array} \right\}$
- $W' = W \cup \begin{cases} before : \langle 0, tReal, ground(can\_do\_a, \chi)\rangle \\ after : \langle tReal, inf, ground(can\_do\_a, \chi)\rangle \end{cases}$

where the new operators $o_a$ and $o_{\neg a}$ both extend $o$. The latter with the delete effect $not\_done\_a$, while $o_a$ extends $o$ with the precondition $can\_do\_a$ and add effect $has\_done\_a$; i.e.:

$$Eff^-_{\dashv}(o_{\neg a}) = Eff^-_{\dashv}(o) \cup \{not\_done\_a\}$$
$$Pre_{\leftrightarrow}(o_a) = Pre_{\leftrightarrow}(o) \cup \{can\_do\_a\}$$
$$Eff^+_{\dashv}(o_a) = Eff^+_{\dashv}(o) \cup \{has\_done\_a\}$$

This ensures that the ground action $a = ground(o_a, \chi)$ must be present in the plan between the times 0 and $tReal$, or $tReal$ and $inf$, depending on the user question, and between those times only. In addition, the user selected action is forced to be performed using the same approach as in Section 4.2. Given the user question above, the HPlan is:

```
0.00: (goto_waypoint kenny wp0 wp2)   [1.45]
1.45: (goto_waypoint_a kenny wp2 wp5) [2.00]
3.45: (goto_waypoint kenny wp5 wp3)   [4.68]
8.13: (goto_waypoint kenny wp3 wp5)   [4.68]
12.81: (goto_waypoint kenny wp5 wp2)  [2.00]
14.81: (goto_waypoint kenny wp2 wp1)  [2.00]
16.81: (goto_waypoint kenny wp1 wp0)  [2.00]
18.81: (goto_waypoint kenny wp0 wp4)  [2.00]
```

### 4.7 Reordering Actions

Given a plan $\phi$, a formal question $Q$ is asked of the form:

*Why is the operator $o$ with parameters $\chi$ used before (after) the operator $n$ with parameters $\chi'$, rather than after (before)? where $o \neq n$ or $\chi \neq \chi'$*

For example, given the example plan in Figure 5 the user might ask:

"Why is $(goto\_waypoint\,kenny\,wp2\,wp1)$ used before $(goto\_waypoint\,kenny\,wp2\,wp5)$, rather than after?"

A user might ask this because there are more connections from $wp5$ than $wp2$. The user might think that if the robot has more choice of where to move to, the planner could make a better choice, giving a more efficient plan.

The compilation to the HModel is performed in the following way. First, a directed-acyclic-graph (DAG) $\langle N, E \rangle$ is built to represent each ordering between actions suggested by the user. For example the ordering of $Q$ is $a \prec b$ where $a = ground(o, \chi)$ and $b = ground(n, \chi')$.

This DAG is then encoded into the model $\Pi$ to create $\Pi'$. For each edge $(a,b) \in E$ two new predicates are added: $ordered_{ab}$ representing that an edge exists between $a$ and $b$ in the DAG, and $traversed_{ab}$ representing that the edge between actions $a$ and $b$ has been traversed.

For each node representing a ground action $a \in N$, the action is disallowed using the compilation from Section 4.3. Also, for each such action a new operator $o_a$ is added to the domain, with the same functionality of the original operator $o$. The arity of the new operator, $arity(o_a)$ is the combined arity of the original operator plus the arity of all of $a$'s sink nodes. Specifically, the HModel $\Pi'$ is:

$$\Pi' = \langle\langle Ps', Vs, As', arity'\rangle, \langle Os, I', G', W\rangle\rangle$$

where:

- $Ps' = Ps \cup \{ordered_{ab}\} \cup \{traversed_{ab}\}, \forall(a,b) \in E$
- $As' = \{o_a\} \cup As, \forall a \in N$
- $arity'(x) = arity(x), \forall x \in arity$
- $arity'(o_a) = arity(o) + \sum_{(a,b)\in E} arity(b), \forall a \in N$
- $arity'(ordered_{ab}) = arity(a) + arity(b), \forall(a,b) \in E$
- $arity'(traversed_{ab}) = arity(b), \forall(a,b) \in E$

- $I' = I \cup ground(ordered_{ab}, \chi + \chi'), \forall(a,b) \in E$, where $\chi$ and $\chi'$ are the parameters of $a$ and $b$, respectively.

In the above, we abuse the $arity$ notation to specify the arity of an action to mean the arity of the operator from which it was ground; e.g. $arity(a) = arity(o)$ where $a = ground(o, \chi)$.

Each new operator $o_a$ extends $o$ with the precondition that all incoming edges must have been traversed, i.e. the source node has been performed. The effects are extended to add that its outgoing edges have been traversed. That is:

$$Pre_\vdash(o_a) = Pre_\vdash(o) \cup \{ordered_{ab} \in Ps', \forall b\}$$
$$\cup \{traversed_{ca} \in Ps', \forall c\}$$
$$Eff_\dashv^+(o_a) = Eff_\dashv^+(o) \cup \{traversed_{ab} \in Ps', \forall b\}$$

This ensures that the ordering the user has selected is maintained within the HPlan.

As the operator $o_a$ has a combined arity of the original operator plus the arity of all of $a$'s sink nodes, there exists a large set of possible ground actions. However, for all $b \in N$, $ordered_{ab}$ is a precondition of $o_a$; and for each edge $(a, b) \in E$ the ground proposition $ground(ordered_{ab}, \chi, \chi')$ is added to the initial state to represent that the edge exists in the DAG. Therefore, the only grounding of the operator that can be performed is the action with parameters $\chi + \chi'$. This drastically reduces the size of the search space.

For example given the user question above, two new operators $node\_goto\_waypoint\_kenny\_wp2\_wp5$ (shown in Figure 7) and $node\_goto\_waypoint\_kenny\_wp2\_wp1$ are added to the domain. These extend operator $goto\_waypoint$ from Figure 2 as described above. The HPlan generated is shown below:

```
(:durative-action
     node_goto_waypoint_kenny_wp2_wp5
  :parameters (?v1 ?v2 - robot
   ?from1 ?to1 ?from2 ?to2 - waypoint)
  :duration ( = ?duration
   (travel_time ?from1 ?to1))
  :condition (and (at start
   (robot_at ?v1 ?from1))
   (over all (connected ?from ?to))
   (at start (ordered_wp2_wp5_wp2_wp1 ?v1
?v2 ?from1 ?to1 ?from2 ?to2)))
  :effect (and (at end (visited ?to1))
   (at start (not (robot_at ?v1 ?from1)))
   (at end (robot_at ?v1 ?to1))
   (at end (traversed_v2_from2_to2 ?v2
?from2 ?to2)) ))
```

Figure 7: An operator added to the original domain to capture an ordering constraint between actions. The operator extends the original $goto\_waypoint$ operator.

```
0.00: (goto_waypoint kenny wp0 wp2)   [1.45]
1.45: (node_goto_waypoint_kenny_wp2_wp5 kenny
kenny wp2 wp5 wp2 wp1) [2.00]
3.45: (goto_waypoint kenny wp5 wp3)   [4.68]
8.13: (goto_waypoint kenny wp3 wp5)   [4.68]
12.81: (goto_waypoint kenny wp5 wp2)  [2.00]
14.81: (node_goto_waypoint_kenny_wp2_wp1 kenny
wp2 wp1) [2.00]
16.81: (goto_waypoint kenny wp1 wp0)  [2.00]
18.81: (goto_waypoint kenny wp0 wp4)  [2.00]
```

# 5    Conclusion

In this paper we have presented an approach to compiling a set of formal contrastive questions into domain independent constraints. These are then used within the XAI paradigm to provide explanations. We have described how these compilations form a part of a series of stages which start with a user question and end with an explanation. This paper formalises and provides examples of these compilations in PDDL 2.1 for temporal and numeric domains and planners.

We have defined a series of questions which we believe a user may have about a plan in a PDDL2.1 setting. These questions cover a large set of scenarios, and can be stacked to create new interesting constraints which may answer a much richer set of questions.

We acknowledge that the questions we provide compilations for do not cover the full set of contrastive questions one may have about a plan. For example the question, "Why is the operator $o$ with parameters $\chi$ used at time $lb < t < ub$, rather than not being used in this time window?", can be answered using a variant of Section 4.5. For future work we plan to investigate which compilations will form an atomic set whose elements can be stacked to cover the full set of possible contrastive questions. We also acknowledge that the compilations we have formalised may have equivalent compilations. However, the ones we have described have proven successful for explanations.

In future work, we will look to extend this work in several ways. While we define how to calculate plans for contrastive cases, we do not take full advantage of contrastive explanations by explaining the *difference* between two plans (Miller 2018). In particular, we will look to extend the presentation beyond just plans into showing the difference between two causal chains as well.

We will explore contrastive explanations with preferences in PDDL 3 (Gerevini and Long 2005).

We will look at producing a language for expressing questions and constraints on plans. LTL will likely play a role in defining the semantics of any such language. Additional concepts concerning plan structure, such as the ability to specify that an action is part of the causal support for a goal or sub-goal, will be needed. As it stands when we add a constraint to include an action, the constraint may be satisfied in trivial ways not relevant to answering the users question. The action may be redundant, or undone in the HPlan as described in (Fox, Long, and Magazzeni 2017). In this case the explanation may not be deemed satisfactory. These additional concepts will help solve this problem, as well as allowing users to ask more expressive questions such as, "Why did you use action A rather than action B for achieving P?".

Finally, we will provide functional and human-behavioural evaluations of our explanations, to assess their effectiveness. To make sure they are both satisfactory from a user perspective, and that they provide actionable insight into the plan.

**Acknowledgements**    This work was partially supported by Innovate UK grant 133549: *Intelligent Situational Awareness Platform*, and by EPSRC grant EP/R033722/1: *Trust in Human-Machine Partnerships*.

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
