# OpenReview forum: "Towards Model-Based Contrastive Explanations for Explainable Planning"
_icaps-conference.org/ICAPS/2019/Workshop/XAIP — XAIP 2019_

### Official Review · AnonReviewer1 · 2019-05-08
**Review 1**

**Rating:** 4
**Confidence:** 3

**Review:**

The paper describes a set of translations and transformations of temporal PDDL problems to support the generation of contrastive explanations.  It outlines how to accomplish this for seven question types related to actions within a temporal plan; for example "why action a instead of action b?" or "why not action a in time interval [0, 2]".

Overall, the paper was straightforward to read and well motivated.  Some minor structural changes could improve the paper (details below).  My main quibble was with the use of arity in the definition.  This was confusing when it was first presented since it was not clear to me why it needed to be included and deviates from the somewhat standard form of planning models.  It wasn't until Section 4.7 that I saw it was possibly necessary, but even then it seemed like this could be left off the descriptions to help improve the paper.   I don't recall any papers that discuss the arity of predicates  -- it seems like most planners will simply refuse to parse such invalid problems.  Also, the formal model starts what I assume was meant to be a lifted form (Ps, Vs..) that are eventually ground (P, V, ..) but I found this confusing.  For both of these, you might consider how to drop these for the sake of clarity, perhaps adding a paragraph to state what assumptions your sweeping with your notation.  Alternatively, more exposition is needed to help the reader get through the introduction of the model.

The approach is strongly reminiscent of plan trajectories from PDDL 3.0, which were used to represent preferences.  This approach fell out of favor after it was shown that soft goals could be compiled away.  However, it occurs to me that further work in this area could benefit from looking at the original PDDL 3.0 specifications as well as the compilations.  There may be some hidden gems in that old work that would apply here.

Other comments:
- What planner was used to generate plans?
- Show Figure 5 first and then show the model.  I started to draw a picture of Figure 3 and then realized I just needed to turn the page.
- Figure, Table, Section, etc., when referring to a single entity (e.g., the only Figure 3 in the paper), are proper nouns and should be capitalized throughout.
- Section 4.5 (forbid) should probably come before 4.4 (delay) since delay uses forbid.  This removes a forward reference.
- the durative-action goto_waypoint_nota should probably be placed in its own figure for consistency.

---

### Official Review · AnonReviewer4 · 2019-05-13
**Highly relevant work; discussion of limitations could be useful; address question language rather than specific questions?**

**Rating:** 3
**Confidence:** 2

**Review:**

The paper follows up on one of Fox et al's proposals for XAIP, answering questions of the type "why does the plan have property A rather than property B?". It assumes the contrastive answer generating a plan with property B for comparison, and it focusses specifically on the sub-problem of designing a compiled planning problem whose solution will be such a plan.

This is completey fine, and highly relevant to XAIP. I think the paper should be accepted.

I have two major comments though:

1. Imho, the approach of generating a plan with property B for comparison has severe limitations. First, the plan may differ from the previous one in arbitrary ways unrelated to A vs B. Second, the plan may satisfy B in trivial ways not relevant to answering the question (e.g., as mentioned by Fox et al, if A and B are applying a particular action in the current state, tjen the new plan may undo action B and use again action A afterwards). These issues do not by any means disvalidate the approach. But I think they need to be at least mentioned.

2. I find it awkward and scientifically unsaisfactory to address this problem as a long list of individual questions. First, this pretends that the qestions have no interreltion, which is clearly not so, isiuble also in the similarities between the compiltions offered. Second, it leaves open all the other questions which might be asked. The authors acknowledge this, but do not offer a solution. How will we ever be able to address the problem in a way that can be reasonably argued to be exhaustive? Why, the rather obvious answer given my own background is to define a question *language* instead of individual questions. Some fragment of logics with temporal elements that allows to state the things the authors want stated here. This would lift the discussion from one about long lists of questions (awkward, infeasible) to one about expressiveness of languages (elegant, compact, standard in CS). This seems rather obvious to me and I wonder why the authors did not consider it, or did not choose to discuss it. Perhaps such an approach would be difficult from an explanation/cognitive point of view? In any case, I do think that at least a brief discussion of this possibility should be given in the paper.

---

### Official Review · AnonReviewer2 · 2019-05-14
**Interesting paper encoding contrastive questions into PDDL2.1**

**Rating:** 4
**Confidence:** 3

**Review:**

This paper deals with the problem of formalising contrastive explanations (i.e., why did you apply action A rather than B?) into PDDL2.1.  As a characteristics, the approach proposed is  domain and planner independent, focusing on temporal planning.  The problem is definitively interesting to the workshop and to the XAIP in general.

The authors propose a set of questions (1..7) that require contrastive explanations. Then, for each question they show how they can be encoded into PDDL2.1 to allow for a contrastive explanations.
The paper is well written and motivated. The contribution is clear and the approach presented is sound, though at preliminary stage, as the authors declared.

I think that using a list of questions is a human-like approach towards explanations in AI Planning, as they allow users understanding the reasons behind the planner's decision through hypotheses, as a human would do. I think this approach - properly extended - is successful. The paper is still at preliminary stage and I think it would benefit from a discussion and presentation at the XAIP workshop.

The user-interaction part to ask questions to users might be beneficial from some recent NLP algorithms, such as word-embedding. It allows one to represent word meaning into a N-dimensional vector space. Words with similar meaning are mapped to a similar position in the vector space. For example, “powerful” and “strong” are close to each other, whereas “powerful” and “Paris” are farther away. The word vector differences also carry meaning. For example, the word vectors can be used to answer analogy questions using simple vector algebra: “King” - “man” + “woman” ≈ “Queen” (Mikolov, Yih, and Zweig, 2013). This knowledge also grows over time, making the system able to "understand" different meaning of sentences on the base of the lexicon used. Ideally, the user might write a question in a natural language, and word-embeddings might be used to "suggest" one of the contrastive questions to be applied.

---

### Decision · Program_Chairs · 2019-05-15

**Decision:**

Accept

**Comment:**

The reviewers agree to accept. Please address all review criticism as best possible for the final paper version and its presentation at the workshop. Looking forward to discuss your work at the workshop!